# Evaluation of in-service training program of laboratory professionals in Amhara Public Health Institute Dessie Branch, northeast Ethiopia: A concurrent mixed-method study

Seid Legesse[1]*, Tefera Alemu[2], Mulugeta Tassew[1], Birtukan Shiferaw[1], Semagn Amare[1], Zerfie Tadesse[1], Minwuyelet Maru[1]

**1** Amhara Public Health Institute Dessie Branch, Dessie, Ethiopia, **2** Amhara Public Health Institute, Bahir Dar, Ethiopia

\* seidmphn2009@gmail.com

## Abstract

### Background

In-service training programs should be evaluated and modified regularly to enhance training quality. However, in Ethiopia, there is no published evidence regarding its effectiveness. Therefore, we evaluated the Amhara Public Health Institute Dessie Branch (APHI_DB) in-service training program using the Kirkpatrick model.

### Methods

In October 2019, a concurrent nested mixed method facility-based cross-sectional study was conducted among 107 laboratory stakeholders from 22 randomly selected government health facilities in the eastern part of the Amhara region. The qualitative part involved inter-views with each of these key stakeholders. We collected data using a semi-structured questionnaire through face-to-face interviews. EpiData 3.1 and Microsoft Excel 2016 software were used for data entry and analysis respectively. The major qualitative findings were narrated and summarized based on four thematic areas to supplement the quantitative findings.

### Results

A total of 107 laboratory personnel were interviewed, which makes a response rate of 97.3%. At the reaction level, 82.1% of the participants agree/strongly agree with the course structures, training contents, and learning tools. Likewise, 85.4% of the participants agreed/strongly agreed on the trainer's knowledge and their communication skills. In addition, 93.1% of the participants stated an improvement in knowledge and skills after attending the training. As a result, 65.6% of them were able to transfer their knowledge and skills into practice. Regarding the training set-ups and environment, 45.1% of the respondents disagree/strongly disagree with the training hall, toilet, cafe, tea and snacks, financial process, and accommodation perdiem.

**Data Availability Statement:** All relevant data are within the paper and its Supporting information files.

**Funding:** We acknowledge Amhara Public Health Institute Dessie Branch for covering data enumeration expense. The funder had no role in study design, data collection and analysis, decision to publish, or preparation of the manuscript.

**Competing interests:** The authors have declared that no competing interests exist.

**Abbreviations:** APHI_DB, Amhara Public Health Institute Dessie Branch.

## Conclusion

Generally, the laboratory in-service training program of APHI_DB was more or less effective. Our findings suggest regular monitoring of each training event and evaluation of training programs against a clearly defined criterion. Furthermore, the institute is mandated to create a conducive learning environment and well-established training setups for trainees.

## Introduction

The in-service training program is one of a commonly used strategy to improve health workers' knowledge, skills, and behaviors on new health approaches and technologies [1, 2]. The ultimate aim of the program is to capacitate health workers with the ability to deliver quality and accessible health services for the entire population [3]. To achieve this objective, a periodic evaluation of a training program should be conducted to identify major program strengths and weaknesses [4]. This helps training organizers to learn from their previous experiences, to know which stages of the training were successful and which not, or whether the approach to the training should be changed or not [5, 6]. The training evaluation also allows the identification of factors that restrain or contribute to a better performance of trained health professionals, suiting the training actions to the organizational needs [7, 8]. After all, the results of the evaluation should be used to modify the contents and conduct of a training program [4].

In the Ethiopian healthcare setting, laboratory professionals perform routine and highly specialized tests to diagnose and/or aid in the treatment of diseases, preventing and solving problems with results, specimens, or instruments, and communicating test results to physicians and/or clinicians [9]. Therefore, they need to be trained on specialized instrumentation and techniques to analyze patients' samples, such as blood, urine, stool, body fluids, tissues, and cells that might show cancer or other diseases [10]. Thus, the Amhara Public Health Institute Dessie Branch (APHI_DB) in collaboration with partners is providing a range of capacity-building training to laboratory and other health professionals in the eastern Amhara region. The training is of various types which primarily focuses on identified priority laboratory needs; which commonly includes specialized laboratory testing services like tuberculosis testing, HIV testing, malaria diagnosis, basic microbiology, viral load determination, gene expert, laboratory quality management system, strengthening laboratory management toward accreditation, safety and bio-security, antiretroviral treatment automation, sample referral linkage and internal audit.

These are independent training courses given for different training participants in separate training sessions. However, most trainees attend over one training type per year. Training contents and curriculum are developed nationally in collaboration with stakeholders and partners and delivered by the Amhara Public Health Institute Dessie Branch in-service training center. We recruit trainees who are working in the medical laboratory setting from public and private health facilities. Then, trainees take part in the short term, face-to-face and single training sessions for a maximum of ten days and return to their workplace. Theoretical sessions are supplemented with practical sessions, videos, case studies, and role-play to maximize a better understanding of the contents. However, in Ethiopia, though health care organizations including APHI_DB spend substantial amounts of time, effort, and funding on health professionals capacity building training, there is no published evidence on the effectiveness of in-service training programs of training centers. Therefore, evaluation of the APHI_DB in-service training program is found to be important, as it can provide information that would alert the plan

for enhancing training effectiveness. In addition, it highlights the major strengths and weaknesses in the training stages and the trainer institute. Thus, we evaluated the overall effectiveness of the APHI_DB in-service training program based on a set of criteria common to the major training types listed above.

## Methods of evaluation

### Evaluation design and period

In October 2019, a concurrent nested mixed method, facility-based cross-sectional study was conducted to assess APHI_DB in-service training program effectiveness.

### Study area and setting

The Amhara Public Health Institute Dessie Branch (APHI_DB) was established in 1982 as a regional laboratory research center and renovated as a branch public health institute in 2016. It is in Dessie town, eastern Amhara, 401 km far from Addis Ababa (the capital city of Ethiopia) to the northeast direction and 490 km from Bahir Dar (the capital city of Amhara regional state). The institute has three dominant entities; namely medical and public health laboratory, research and technology transfer, and public health emergency management [11].

The laboratory wing provides technical support to health facilities found in the six zones, provides and serves as an in-service training center for various capacity-building trainings, referral and specialized laboratory test sites, performs public health laboratories for epidemic-prone disease and conducts an external quality assessment, laboratory mentorship and accreditation. The institute has a mandate to establish and maintain a high-quality and sustainable laboratory system throughout the sub-region. It also delivers quality and accessible laboratory services related to the occurrence, causes, prevention and diagnosis of major diseases of public health importance and to establish and support National Laboratory Quality Assurance programs and systems [12].

To realize its mission, the institute benefits from the generous support of the Ethiopian and regional governments and various national and international organizations. Some organizations that support the institute include the World Health Organization through its Global Fund, Centre for Disease Control, Japan International Cooperation Agency, International Center for AIDS Care and Treatment Programs (I-CAP) and various non-governmental organizations through joint projects.

### Sample size and sampling technique

First, out of 30 external quality assurance health facilities, 22 were randomly selected to represent the eastern part of the region. Then, nearly five laboratory personnel from each government health facility were randomly selected, making a final sample size of 107 respondents. The qualitative part also involved interviews with each of these key stakeholders. All participants were laboratory personnel with a minimum of one year of work experience and who had ever received training in Amhara Public Health Institute Dessie Branch within the past two years.

### Data collection

A total of 107 laboratory stakeholders were interviewed in person at their respective health facilities using a semi-structured questionnaire. We also included all these stakeholders in the qualitative key informant interviews. The questionnaire had both open-ended and closed-ended questions. Respondent's opinions, suggestions and comments were received through

key stakeholder interviews. The questionnaire covered information on laboratory training profiles, trainee's reactions towards the course structure, training content and training tools, trainer's knowledge and communication skills, new knowledge and skills gained, change in job behavior and performance, training set-ups and environment. Our respondents were laboratory personnel from selected health centers and hospitals. The data were collected by the investigators, considering that they were not assigned as a trainer and not from laboratory directorate.

## Data quality assurance

To minimize the subjectivity of responses from stakeholders, the investigators themselves took part as data collectors. During the session of each visit, we briefed the respondents about the purpose of the assessment, which was to evaluate the effectiveness of the in-service training program of APHI_DB; not merely the individual's performances. Besides, respondents were asked to feel open and comfortable to give their opinions, suggestions and comments that can help to improve in-service training programs and future laboratory services.

## Data analysis

We graded all responses as 'strongly agree, agree, not to decide, disagree, strongly disagree and yes or no, except for the open-ended questions. Data completeness was checked daily and the quantitative data entry was done through EpiData 3.1 software and exported to Microsoft Excel 2016 for analysis. Data analysis was done at four levels, i.e. (i) Trainees reaction towards training, (ii) Trainers knowledge on the course & their communication skills, (iii) Learning of training contents, (iii) Changes in job behavior after attending training (iv) Training setups, benefits and financial issues. We computed descriptive statistics to describe the data. Then, we narrated the major qualitative findings and summarized based on the above four thematic areas to supplement the quantitative findings.

## Ethical approval and consent to participate

Ethical approval was obtained from the Institutional Review Committee of Amhara Public Health Institute. A formal letter was given to each health facility to get permission and cooperation. After briefing the purpose of the study, oral consent was obtained from each participant.

## Operational definitions

**In-service training.**   A range of short term, practical training courses given by APHI_DB for laboratory professionals working at health facilities to scaling up their knowledge and skills.

**Training effectiveness.**   We evaluated the overall effectiveness of APHI_DB in-service training program using criteria common to all training types (Table 2) given by the institute. For each level of evaluation, we judge effectiveness as not effective when average values fall less than $50^{th}$ percentile, partially effective between $50^{th}$ and $75^{th}$ percentile and effective above $75^{th}$ percentile. But, in Table 8, we used the term adequate, satisfactory and appropriate instead of the word effective.

## Results

### Socio-demographic characteristics of respondents

A total of 107 laboratory personnel who are working in 22 health facilities (18 hospitals and 4 health centers) were interviewed with a response rate of 97.3 percent. The proportion of male

respondents was 60.7%. The age of the participants ranged from 20 to 49 years; 42% were between 25 and 29 years old. Most participants are laboratory technicians (53.3%), while 44% have Bachelor of Science degree in a medical laboratory. Regarding respondent's work experience, 44% have one up to three years' experience whereas 34.6% of them had over five years of work experience and the rest 21.5% are in between three and five-year experiences. The majority (84.1%) of the respondents are working in the hospital and the rest 15.9% are in health centers (Table 1).

## Distribution of in-service training among respondents

As depicted in Table 2, about 53.3% of laboratory personnel were trained on the new national HIV testing algorism, 52.3% on tuberculosis fluorescent microscopy and Ziehl-Neelsen diagnosis, 69.2% on malaria diagnosis, 26.2% on gene expert and 69.2% on laboratory quality management system. Only 8.5%, and 6.5% of laboratory personnel were trained on dry blood spot/ viral load sample collection and basic microbiology courses respectively.

The qualitative findings showed that training opportunities were almost equally spread across the health facilities and the number of trained personnel seems adequate. However, few of the respondents believed that there was trainee selection bias or selection criteria are not fair from facility, to facility and also the calling modality was not appropriate, particularly at the zonal health department level.

## The satisfaction of trainees with the course structures, training contents and tools

In this theme, 65.4% of the study participants mentioned that training contents were in depth enough, whereas 29.9% had a reservation on this. Another 85% also stated that learning aids/ training tools assisted their learning sessions. Most of the respondents (79.4%) agree/strongly agree that technology/lab equipment was working properly during the training time and able to provide all the practices for the course, though 17.8% of the respondents strongly disagree/ disagree on this idea. The same proportions of respondents (80.4%) agree/strongly agree that

**Table 1. Sociodemographic characteristics of respondents, northeast Ethiopia, October 2019 (N = 107).**

| Variables | Category | Number (%) |
|---|---|---|
| Sex | Male | 65 (60.7) |
| | Female | 42 (39.3) |
| Age category | 18–24 years | 38 (35.5) |
| | 25–29 years | 45 (42.1) |
| | 30–34 years | 15 (14) |
| | >= 35 years | 9 (8.4) |
| Work experience | 1–3 years | 47 (43.9) |
| | 3–5 years | 23 (21.5) |
| | >5 years | 37 (34.6) |
| Level of education | Diploma | 57 (53.3) |
| | Degree | 47 (43.9) |
| | Masters | 3 (2.8) |
| Health facilities involved | Hospitals | 18 (81.8) |
| | Health centers | 4 (18.2) |
| **Respondents'** workplace | Hospital | 90 (84.1) |
| | Health center | 17 (15.9) |

**Table 2. In-service training types received by respondents from Amhara Public Health Institute Dessie Branch, northeast Ethiopia, October 2019 (N = 107).**

| Type of training | Received | Frequency (%) |
|---|---|---|
| New HIV testing algorism | Yes | 49(45.8) |
| Tuberculosis Fluorescent Microscopy & Ziehl-Neelsen diagnosis | Yes | 56(52.3) |
| Malaria diagnosis | Yes | 74(69.2) |
| Laboratory Quality Management System | Yes | 57(53.3) |
| Strengthening Laboratory Management Toward Accreditation | Yes | 47(43.9) |
| Safety and Bio-security | Yes | 43(40.2) |
| Anti-retroviral treatment automation | Yes | 54(50.5) |
| Dried Blood Spots specimen for HIV viral load sample collection | Yes | 9(8.4) |
| Sample Referral Linkage | Yes | 27(25.2) |
| Basic Microbiology | Yes | 7(6.5) |
| Gene Expert | Yes | 28(26.2) |
| Internal Audit | Yes | 20(18.7%) |
| Compassion, Respectful and Caring Professional | Yes | 12(11.2) |

the course provided opportunities to practice and would improve their job performance. Besides, most respondents (86.9%) agree/strongly agree that the outcome of the courses was successfully achieved compared to meet their general expectations of the training and the courses gave trainees a clear understanding of the goals and objectives of the training before they started the training (96.3%) (Table 3).

In agreement with the quantitative finding, most of the respondents in the qualitative in-depth interview believed that the course structure, content and training tools of the training were clear and understandable; courses providing opportunities to practice and the contents were in-depth enough. Similarly, the availability of laboratory equipment and tools has been appreciated, especially in Strengthening Laboratory Management Towards Accreditation and Automation training. Some of the respondents risen "too short practice sessions and shortage of necessary machines and equipment" as a problem regarding course structure, contents and training tools.

**Table 3. Distribution of responses on the course structures, training contents and tools of training in Amhara Public Health Institute Dessie Branch, northeast Ethiopia, October/2019 (N = 107).**

| Survey items | Strongly agree or agree (%) | Strongly disagree or disagree (%) | Not to decide |
|---|---|---|---|
| Goals and objectives clearly stated before you started the training? | 103 (96.3) | 4(3.7) | 0 |
| Training length was sufficient to deliver the course? | 87(81.3) | 20(18.7) | 0 |
| Learning aids assisted in your training? | 91 (85) | 13(12.1) | 3(2.8) |
| Technology/lab equipment was working properly during the training or practical session? | 85 (79.4) | 19(17.8) | 3(2.8 |
| Was the content in-depth enough? | 70 (65.4) | 32(29.9) | 5(4.7) |
| Was the course provided opportunities to practice? | 86 (80.4) | 20(18.7) | 1(0.9) |
| Was over one training style used or was conducive to my learning style? | 88(82.2) | 15(14) | 4(3.7) |
| Course outcome was successful compared to meet their expectations? | 93(86.9) | 13(12.1) | 1(0.9) |
| Cumulative satisfaction score | 82.1% | 15.9% | 2% |

**Table 4. Classifications of trainer's knowledge and skills in Amhara Public Health Institute Dessie Branch, northeast Ethiopia, October/2019 (N = 107).**

| Survey items | Strongly agree or agree (%) | Not to decide | Strongly disagree or disagree (%) |
|---|---|---|---|
| Trainer's delivery skills | 97(91) | 7(6.5) | 3(2.8) |
| Participatory and interactive | 79 (73.8) | 13(12.2) | 15(14) |
| Trainer's knowledgeable | 92 (86) | 9(8.4) | 6(5.6) |
| Trainer's communication skills | 95 (88.8) | 6(5.6) | 6(5.6) |
| Trainer responsive to questions | 96(89.7) | 9(8.4) | 2(1.87) |
| Trainer preparedness for class | 89(83.2) | 10(9.35) | 8(7.5) |
| Cumulative score | 85.4% | 8.4% | 6.3% |

### Trainer's knowledge and skills during training the course

The proportions of respondents that strongly agree/agree on the trainer's knowledge about the course contents, their communication and delivery skills were nearly similar. Hence, 91% of respondents strongly agree or agree that trainers be able to deliver training courses properly and most respondents (88.8%) strongly agree or agree that trainer's communication skills were able to provide all the information needed for the course. About 86% of respondents strongly agree or agree regarding the trainer's knowledge of the course. Yet, 13.7% of respondents strongly disagree or disagree with that of the trainer's ability to create a participatory and interactive training environment; though 12% of the respondents cannot decide on this issue (Table 4).

Findings from the qualitative part also revealed that most respondents acknowledged the trainer's knowledge, communication and delivery skills. Others also reported more positive notions of knowledge, skills, and willingness to train based on their praising. In contrary to these, as stated by a few of the respondents, some problems related to knowledge and skills of trainers includes; undermining of a trainee, a few trainers are aggressive, sometimes trainers are none expertise or non-laboratory professionals, few trainers presentation styles is lecture so better to be entertaining style, trainers did not take the training of trainers.

### New knowledge and skills gained in the training (learning the contents)

In this stage, 93.1% of the trainees strongly agree/agree that their knowledge and skills have improved because of having attended the training. Likewise, 92.5% of them strongly agree/agree that the practical sessions of the training have improved their skills and professional competencies (Table 5).

### Change in job behavior and performance in the workplace (behavioral level)

At this level of evaluation, 93.5% of trainees mentioned as if they are applying the knowledge and skills they gained during the training in their workplace, and 69.2% of them tried to qualify other laboratory personnel's in their working areas, 68.2% of them are using the training

**Table 5. Improvement in trainee's knowledge and skills after attending training in Amhara Public Health Institute Dessie Branch, northeast Ethiopia, October/2019 (N = 107).**

| Item description | Strongly agree | agree/ | Not to decide | Disagree |
|---|---|---|---|---|
| Do you feel that your knowledge or skills have improved by taking the training | 37(34.6) | 63(58.9) | 3(2.8) | 4(3.7) |
| Do you believe that the practical exercises were good and improved your skill | 35(32.7) | 64(59.8) | 4(3.7) | 4(3.7) |
| Average item score | 33.7% | 59.4% | 3.3% | 3.7% |

**Table 6. Trainees knowledge and skill transfer into practice in their working facility, northeast Ethiopia, October/2019 (N = 107).**

| Variables | Yes | |
|---|---|---|
| | Number | Percent |
| Were the learned knowledge and gained skills used in the workplace? | 100 | 93.5 |
| Are the training materials on use/shared with other staff in your workplace? | 73 | 68.2 |
| Assign trained personnel's on proper work position | 48 | 44.9 |
| Trying to qualify other health professionals in a workplace | 74 | 69.2 |
| Would you recommend this training to a colleague | 88 | 82.2 |
| Would you consider further training on the topic on your own | 91 | 85 |
| Is there a registration system that shows trained staffs in the lab unit | 28 | 26.2 |
| Identified thematic area for training | 60 | 56.1 |
| Do you have a monitoring and evaluation system | 68 | 63.6 |
| Organization announce when trained professional change work site | 72 | 67.3 |
| Average item score | | 65.6% |

materials properly. However, 73.8% of the respondents replied no registration system that shows trained staff in the lab unit of their facility, and 55.1% of them also showed that they didn't assign trained persons to the appropriate positions. On average, as stated in Table 6 below, 65.6% of the trainee's transferred their knowledge and skill they gained during the training into practice.

## Training setups and environment

In this theme, more than half of the respondents (51, 4%) disagree/strongly disagree with the suitability of training setups and environment (training hall, toilet, and cafe) for training. In the meantime, 46.7% of respondents strongly disagree/disagree with the accommodation perdeim and/or refreshments of the training program. Also, 28.9% of the respondents mentioned as if the calling modality for the training was inappropriate. Still, 69.2% of the respondents strongly agree or agree on the time and season of the training (Table 7).

The finding from the qualitative part also revealed that many of the respondents had a complaint on training setups, benefits and financing processes. Thus many of them have been fill discomfort with the training hall, toilet, cafe, the financing process, insufficiency of perdiem, and not happy with tea and snacks. On top of these, some respondents have been disgusted by the extended waiting time, guilty words and insults to receive their accommodation perdiem.

**Table 7. Respondents' description of training set-ups, benefits and finance-related factors in Amhara Public Health Institute Dessie Branch, northeast Ethiopia, October/2019 (N = 107).**

| Item description | Response grading | | | | |
|---|---|---|---|---|---|
| | Strongly /agree | agree | Not to decide | Disagree | Strongly disagree |
| Training facilities/hall, toilet, cafe,…) was suitable for training | 4(3.7) | 26(24.3) | 22(20.6) | 38(35.5) | 17(15.9) |
| Adequacy of accommodation perdiem/tea breaks for the training | 12(11.2) | 20(18.7) | 25(23.4) | 40(37.4) | 10(9.3) |
| Was training time and season appropriate? | 17(15.9) | 57(53.3) | 17(15.9) | 14(13.1) | 2(1.9) |
| The way of calling was appropriate | 19(17.9) | 57(53.3) | 0 | 12(11.2) | 19(17.7) |
| Training call was heard in an appropriate time | 17(15.9) | 65(60.7) | 19(17.8) | 6(5.6) | 0 |
| Average item score | 12.9% | 42% | 15.5% | 20.6% | 9% |

**Table 8. A summary table of the effectiveness status of major evaluation themes in APHI_DB, northeast Ethiopia.**

| Major themes /Level of evaluation | No. of indicators | Cumulative score derived from the data (%) | | Judgment level |
|---|---|---|---|---|
| | | Strongly agree/agree | Strongly disagree/disagree | |
| The reaction of trainees towards training (Reaction level) | 8 | 82.1 | 15.9 | Satisfactory |
| Trainer's knowledge and skills | 6 | 85.4 | 6.3 | Adequate |
| New knowledge and skills acquired (Learning level) | 2 | 93.1 | 3.7 | Adequate |
| Change in job behavior and performance (Behavioral level) | 10 | 65.6 | 34.4 | Less satisfactory |
| Training setups & environment | 5 | 54.9 | 29.6 | inappropriate |
| Average item score | 31 | 76.2% | 18% | Effective |

Overall, the authors have tried to summarize the effectiveness status of each level of training evaluations. As a result, participant's reaction and their learning events were rated as satisfactory and adequate, respectively; while post-training behavioral change and the training setups were less satisfactory and inappropriate respectively. In general, the average cumulative scores of each level of evaluations fall in the fourth quartile (76.2%). Thus, as per our operational definition, this figure shows as the in-service training program of APHI_DB was more or less effective (Table 8).

## Discussion

This study has tried to assess the effectiveness of the Amhara Public Health Institute Dessie Branch in-service training program using the Kirkpatrick models of training evaluation. Thus, in the first theme of evaluation, most participants were motivated and reacted positively to the course structures, contents and training tools which shows excellences in pre-training phase like training module preparation, method selection, training need assessment and trainee recruitment. Similar finding is also documented in a study from Egypt [13].

Regarding the trainer's knowledge and skills on the course content, most participants point out that the trainer's knowledge about the course content, their good communication and delivery skills were adequate to provide all the information needed for the course. This shows the availability of an experienced trainer that has received "training of trainers" on each training type. Besides, allocating enough trainers with sufficient training time will enable trainers to entertain novel ideas in a room which provides more discussion opportunities to trainees to grasp all the knowledge and skill competencies of the training and in depth exchange of best practices and challenges. Conversely, the allocation of non-competent trainers in the present study could be attributed to trainer's knowledge gap and/or lack of expertise or intentionally assigned by training organizers; just for the seek of individual benefits or because of lack of experienced trainers. These finding shows a weakness in providing training to address actual training gaps. This kind of problem could be resolved by allocating most senior and experienced trainers rather than the mere allocation of trainers based on some other criteria.

In the second level of our evaluation, most participants mentioned that they have refreshed and/or gained new knowledge and skills during the training. In agreement with this, a study done in Sweden and Sri Lankan also pointed out the effectiveness of in-service training to equip trainees with the new knowledge and skills required for the job [8, 14]. Another systematic review emphasized the importance of training to health professionals to enrich their pre-existing knowledge and skills [15]. Our finding at this level implies the effectiveness of short term in-service training to strengthen health workers' capacity on new health approaches or technology for improved health service provisions. This finding for in-service training

program holders suggests the importance of improving in-service training coordination, planning, curriculum development, provision, evaluation and accreditation.

Our evaluation at the behavioral level asserted that the knowledge and skills they gained during the training have positively affected their work performance in most of the cases. They also evidenced this result in a recent systematic review of essential newborn care training that suggests improvements in delivery room newborn care practices as compared with usual care [16]. In this phase, our finding implies the success of APHI_DB training program to achieve its goal deeming that success at a behavioral level will lead to a result. Yet, trainee's implementation effort and their competences should be further strengthened through supportive supervision to facilitate the outcome of the training.

Nearly half of the trainees declared that the training setups and environments were not suitable and appropriate for learning. Thus, many of them have been fill discomfort with the training halls, toilets, cafe, the financing process and insufficiency of perdiem, and not happy with tea and snack refreshments. These could be partially explained because the institute is currently working in temporary and non-standardized building blocks, with confined and limited classrooms. Besides, the daily perdiem might not be enough to accommodate trainee's daily expenses in Dessie city, which disturbs trainee's attention and mind during their stay in the training. Unless otherwise these kinds of problems are identified and resolved early, they will grow and amplify themselves to the extent of causing a significant negative impact on the training program and organizational achievement [4].

A limitation of the present study includes the lack of an observational part of the trainee's practice and performance in their workplace that usually affects the quality of laboratory tests. Trainees' knowledge and skill improvement are measured subjectively based on trainees' responses without pre and post-test results. Also, this study did not address the result level of Kirkpatrick model that investigates the total training costs against its benefits. This shows the need to study further at a deeper level to get a holistic picture of APHI_DB training program effectiveness.

## Conclusions

Generally, the laboratory in-service training program in Amhara Public Health Institute Dessie Branch was effective with its limitations. To meet its overall organizational aim, the institute should regularly monitor and evaluate the effectiveness of its in-service training program based on a pre-defined set of performance criteria. Training organizers should use lessons learned /feedbacks in the present study to create a conducive training environment for learning which improves the quality of training services.

## Supporting information

**S1 Appendix. This is the tool used for data collection.**
(DOCX)

## Acknowledgments

We acknowledge Amhara Public Health Institute Dessie Branch for its support in conducting this study. We also appreciate the study participants for their willingness to take part in the study.

## Author Contributions

**Conceptualization:** Seid Legesse, Tefera Alemu, Birtukan Shiferaw, Minwuyelet Maru.

**Data curation:** Seid Legesse, Mulugeta Tassew, Birtukan Shiferaw, Zerfie Tadesse.

**Formal analysis:** Seid Legesse, Tefera Alemu, Mulugeta Tassew.

**Investigation:** Seid Legesse, Tefera Alemu, Mulugeta Tassew, Birtukan Shiferaw, Zerfie Tadesse.

**Methodology:** Seid Legesse, Tefera Alemu, Minwuyelet Maru.

**Supervision:** Seid Legesse, Minwuyelet Maru.

**Validation:** Seid Legesse, Tefera Alemu.

**Visualization:** Seid Legesse, Tefera Alemu, Mulugeta Tassew, Semagn Amare.

**Writing – original draft:** Seid Legesse, Tefera Alemu, Semagn Amare.

**Writing – review & editing:** Tefera Alemu, Semagn Amare.

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
