## [Decision Letter · Decision Letter 0]

6 Aug 2020

PONE-D-20-10999

Evaluating the effectiveness of in-service training program of Amhara Public Health Institute Dessie Branch, Northeast Ethiopia: a concurrent nested mixed quantitative/qualitative facility based cross sectional study

PLOS ONE

Dear Dr. Hassen,

Thank you for submitting your manuscript to PLOS ONE. After careful consideration, we feel that it has merit but does not fully meet PLOS ONE’s publication criteria as it currently stands. Therefore, we invite you to submit a revised version of the manuscript that addresses the points raised during the review process.

Please address my comments as well as those made by reviewer 1.

We look forward to receiving your revised manuscript.

Kind regards,

Conor Gilligan

Academic Editor

PLOS ONE

Additional Editor Comments:

Thank you for submitting this paper to PLOS ONE. It is certainly worthwhile to scrutinise and evaluate in-service training programs, particularly in low and middle income countries, so i commend you on this effort.

The paper needs some re-working before it can be accepted for publication. Please see reviewer 1's comments as well as my own suggestions below.

1. The context of the training is not clear - are these standard training programs delivered by a national agency or varied programs delivered by a range of groups? The authors do mention the training institution but this could be described in more detail in the introduction to provide context. Further, the objectives and nature of training could be described - the results detail the topics and skills covered but it is not clear if the training is a single course designed to cover all these topics, a range of courses covering separate topics, or something else? Do participants select courses based on the topics covered/time of offering etc or are they directed to attend? Do participants attend more than one course? do workplaces support their employees to attend...etc etc. Please provide this context in the introduction.

2. The introduction talks about both laboratory inservice and quality of care improvement training but it seems that the study relates to the former only - again, please clarify the context and objectives of this study and the training being evaluated.

3. More detail is needed to describe the survey and qualitative data collection methods. There is a reference to 'visits' - was data collected in person? How many people completed the survey, and how many participated in interviews?

4. The results could be presented more succinctly - the tables are unnecessarily detailed (e.g. you don't need the frequency for both yeas and no responses). Results could be more meaningful if they were linked to the objectives of the training, and you could also link the outcomes with participants' level of experience etc.

4. Please provide more detail about the thematic analysis methods.

5. The discussion and conclusions could be improved by providing a more clear direction in terms of future directions and recommendations for improvement of training programs.

6. Please engage an English language editor or native English speaker to assist with improving the grammar and writing.

Journal Requirements:

4.Thank you for stating the following in the Acknowledgments Section of your manuscript:

"We acknowledge Amhara Public Health Institute Dessie Branch for covering data enumeration expense."

6. Please amend your list of authors on the manuscript to ensure that each author is linked to an affiliation. Authors’ affiliations should reflect the institution where the work was done (if authors moved subsequently, you can also list the new affiliation stating “current affiliation:….” as necessary).

7. We note you have included tables to which you do not refer in the text of your manuscript. Please ensure that you refer to Tables 3. 4, 5, 6, 7 in your text; if accepted, production will need this reference to link the reader to the Tables.

8. Please upload a copy of Supporting Information which you refer to in your text on page 15 (Availability of data and materials section).

Reviewers' comments:

Reviewer's Responses to Questions

**Comments to the Author**

1. Is the manuscript technically sound, and do the data support the conclusions?

Reviewer #1: Partly

Reviewer #2: Yes

2. Has the statistical analysis been performed appropriately and rigorously? 

Reviewer #1: No

Reviewer #2: Yes

3. Have the authors made all data underlying the findings in their manuscript fully available?

Reviewer #1: Yes

Reviewer #2: Yes

4. Is the manuscript presented in an intelligible fashion and written in standard English?

Reviewer #1: No

Reviewer #2: Yes

5. Review Comments to the Author

Reviewer #1: This is a very important paper and very relevant to improve in-service training in low- and middle-income countries. I agree with the authors that such outcome assessment do identify training gaps to improve. However, the paper is a lot limited which can be resolved. Below are some of the areas that need the authors’ attention.

1. The title of the paper is too long. Can you please consider a shorter but attention grapping title?

2. I do understand that the authors may not be native English speakers, the paper is filled with grammatical errors and inconsistencies.

3. Authors indicated that “…. only 65.6% of participants transferred their knowledge and skills in to practice”. This is a major outcome measure, but the authors failed to indicate how it was measured.

4. The introduction is inconsistent and difficult to follow. It starts with objectives, then to needs, gaps, and then back to objectives. This confuses the readers to know which of the objectives the authors are out address.

5. It is clear that the investigators collected the data themselves. What is not very clear is if they were the trainers? If they were, the results may be skewed because the trainees (in this case the respondents) will want to please the researchers especially in a qualitative design.

6. There are a lot of important points made in the results section, but some were never followed through in the discussion section. If any issue raised in the results section that the authors feel not relevant to discuss, should be deleted.

7. Authors should avoid blank statements that suggest a pre- and post-training assessments. This is evident in third paragraph of the discussion section.

Reviewer #2: Thanks for giving the opportunity to review this manuscript. Congratulations to authors for conducting this manuscript.

This manuscript has been well written. I have no further comments. I would like to request you

6. PLOS authors have the option to publish the peer review history of their article (what does this mean?). If published, this will include your full peer review and any attached files.

Reviewer #1: No

Reviewer #2: No

---

## [Author Response · Author response to Decision Letter 0]

30 Aug 2020

Dear PLOS ONE 

 Her is authors point by point response letter for the comments raised by each of the reviewers/editors. The authors would like to thank the reviewers for their valuable comments. Care has been taken to improve the work and address their concerns as per the specific comments below.

Editor #1. 

Comment 1: 1. The context of the training is not clear - are these standard training programs delivered by a national agency or varied programs delivered by a range of groups? The authors do mention the training institution but this could be described in more detail in the introduction to provide context. Further, the objectives and nature of training could be described - the results detail the topics and skills covered but it is not clear if the training is a single course designed to cover all these topics, a range of courses covering separate topics, or something else? Do participants select courses based on the topics covered/time of offering etc or are they directed to attend? Do participants attend more than one course? do workplaces support their employees to attend...etc etc. Please provide this context in the introduction.

Response 1: Comment accepted and incorporated. You can see under paragraph 2 and 3 of the introduction section.

Comment 2: 2. The introduction talks about both laboratory inservice and quality of care improvement training but it seems that the study relates to the former only - again, please clarify the context and objectives of this study and the training being evaluated.

Response 2: Comment accepted and modified. The objective of this study is to evaluate the effectiveness of in service training program of Amhara public health institute that delivers different capacity building trainings for laboratory professionals working under government health facilities. Therefore, this study did not evaluate a single training type; rather it has tried to evaluate the overall in service training program effectiveness based on predefined evaluation criteria that are common to most training types. 

Comment 3: 3. More detail is needed to describe the survey and qualitative data collection methods. There is a reference to 'visits' - was data collected in person? How many people completed the survey, and how many participated in interviews?

Response 3: A total of 107 key informants/stakeholders were interviewed in the qualitative data collection. The qualitative questions are embedded in to the quantitative parts. The data is collected through face to face interviews. This is explained in the method section line number………

Comment 4: 4. The results could be presented more succinctly - the tables are unnecessarily detailed (e.g. you don't need the frequency for both yeas and no responses). Results could be more meaningful if they were linked to the objectives of the training, and you could also link the outcomes with participants' level of experience etc.

Response: 4: Comment accepted. Actually most of our tables responses are categorized based on likert scale which is mandatory to present all responses in a table. It is only table 2 and 6 that are classified as yes or no response. We believe that it seems a missing data if we omit either of yes or no response.

Comment 5: 4. Please provide more detail about the thematic analysis methods.

Response 5: Also the qualitative data was narrated and summarized based on four thematic areas, i.e. (i) Trainees reaction towards training, (ii) Trainers knowledge on the course & their communication skills, (iii) Learning of training content, (iii) Changes in job behavior after attending training (iv)Training set ups, benefits and financial issues. See under data analysis section.

Comment 6: 5. The discussion and conclusions could be improved by providing a more clear direction in terms of future directions and recommendations for improvement of training programs.

Response 6: Comment accepted. We revised based on comments. See the track changes under discussion section. 

Comment 8: 1. Please ensure that your manuscript meets PLOS ONE's style requirements, including those for file naming.

Response 8: Comment accepted and revised according to the journal guideline.

Comment 9: 3. Please include additional information regarding the survey or questionnaire used in the study and ensure that you have provided sufficient details that others could replicate the analyses. For instance, if you developed a questionnaire as part of this study and it is not under a copyright more restrictive than CC-BY, please include a copy, in both the original language and English, as Supporting Information.

Response 9: Comment accepted and evaluation tool is uploaded as supporting information file. 

Comment 10: 4. We note that you have provided funding information that is not currently declared in your Funding Statement. However, funding information should not appear in the Acknowledgments section or other areas of your manuscript. We will only publish funding information present in the Funding Statement section of the online submission form.

Please remove any funding-related text from the manuscript and let us know how you would like to update your Funding Statement. 

Response 10: Comment accepted and funding related statement is deleted from the acknowledgement section.

 Comment 11: 6. Please amend your list of authors on the manuscript to ensure that each author is linked to an affiliation. Authors’ affiliations should reflect the institution where the work was done (if authors moved subsequently, you can also list the new affiliation stating “current affiliation:….” as necessary).

Response 11: Comment accepted and corrected.

Comment 12: 7. We note you have included tables to which you do not refer in the text of your manuscript. Please ensure that you refer to Tables 3. 4, 5, 6, 7 in your text; if accepted, production will need this reference to link the reader to the Tables.

 Response 13: Comment accepted and tables are cited in the manuscript.

Comment 14: 8. Please upload a copy of Supporting Information which you refer to in your text on page 15 (Availability of data and materials section).

[Response 14: Comment aaccepted. Data collection tool is uploaded as supporting information 1.

Reviewers' comments:

Reviewer #1

Comment 1: 1. The title of the paper is too long. Can you please consider a shorter but attention grapping title?

Response 1: Comment accepted and title is modified.

Comment 2: 2. I do understand that the authors may not be native English speakers, the paper is filled with grammatical errors and inconsistencies.

Response 2: Comment accepted. The manuscript is edited with someone fluent in English.

Comment 3; 3. Authors indicated that “…. only 65.6% of participants transferred their knowledge and skills in to practice”. This is a major outcome measure, but the authors failed to indicate how it was measured.

Response 3; A total of 10 questions were used to measure trainees change in job behavior and performance in their workplace and the average is found to be 65.6%( see table 6).

Comment 4: 4. The introduction is inconsistent and difficult to follow. It starts with objectives, then to needs, gaps, and then back to objectives. This confuses the readers to know which of the objectives the authors are out address.

Response 4; Comment accepted. We have tried to rewrite the introduction part to keep ideas coherent. You can see the track change version of the manuscript.

Comment 5: 5. It is clear that the investigators collected the data themselves. What is not very clear is if they were the trainers? If they were, the results may be skewed because the trainees (in this case the respondents) will want to please the researchers especially in a qualitative design.

Response 5: The investigators/data collectors are from research and technology transfer, public health emergency, monitoring and evaluation and human resource directorates, and none of them are from laboratory directorate. Therefore, none of them are mandated to provide trainings for laboratory professionals.

Comment 6. Authors should avoid blank statements that suggest a pre- and post-training assessments. This is evident in third paragraph of the discussion section.

Response 6: Comment accepted and modified accordingly. See paragraph three under discussion section.

---

## [Decision Letter · Decision Letter 1]

29 Sep 2020

PONE-D-20-10999R1

In-service training program evaluation of Amhara Public Health Institute Dessie Branch, northeast Ethiopia: a concurrent mixed-method study

PLOS ONE

Dear Dr. Hassen,

Thank you for submitting your manuscript to PLOS ONE. After careful consideration, we feel that it has merit but does not fully meet PLOS ONE’s publication criteria as it currently stands. Therefore, we invite you to submit a revised version of the manuscript that addresses the points raised during the review process.

We look forward to receiving your revised manuscript.

Kind regards,

Conor Gilligan

Academic Editor

PLOS ONE

Additional Editor Comments (if provided):

The paper is much improved and almost ready to accept but it does require some further minor corrections and careful English editing.

Line 29 – suggest ‘involved interviews with each of these key stakeholders’

Line 34 –The abstract needs some re-working for language. The results are presented as ‘x% agree/strongly agree with…’ doesn’t fit grammatically. I suggesting changing to “…agree/strongly agree that the trainers had adequate knowledge and skills….” Etc etc

Line 50 – replace could with should

Line 109 – here it says 110 respondents – then in line 113 it says 107 - what happened to 3 participants?/which number is correct?

Line 113 – add ‘in-person’ and the location in which the interviews occurred

Line 115 ‘close’ should be ‘closed’

Lines 152 and 153 – I am unclear what you mean by diploma holders vs first degree professionals – can you please explain the qualifications of respondents?

Line 164 – should be algorithm?

Table 2 – please just show the yes results. Being dichotomous, the ‘no’s’ can be assumed from this

Line 205 – ‘as a potential to confident to the trainers’ – this needs to be re-worded

Lines 220 – 227 – this section needs some elaboration to better describe the issues being raised

Line 231 – “Agreeing with the training set-up” doesn’t make sense grammatically – was the question that it was appropriate/adequate/conducive to learning…please clarify

Table 8 talks about the ‘effectiveness’ of different elements – I suggest using ‘effective’ where describing the learning and transfer of knowledge to the workplace, but ‘appropriate’ or ‘adequate’ may be better for other elements such as the training set up

I think the findings need to more clearly inform conclusions/recommendations including the need for more experiential/practical learning. Obviously the adequacy of venues etc should also be considered.

I was also interested in the comment about the judgment regarding access to perdiem – I think this needs some exploration as it is unclear why this would occur and in what context.

Reviewers' comments:

Reviewer's Responses to Questions

**Comments to the Author**

1. If the authors have adequately addressed your comments raised in a previous round of review and you feel that this manuscript is now acceptable for publication, you may indicate that here to bypass the “Comments to the Author” section, enter your conflict of interest statement in the “Confidential to Editor” section, and submit your "Accept" recommendation.

Reviewer #1: All comments have been addressed

2. Is the manuscript technically sound, and do the data support the conclusions?

Reviewer #1: Yes

3. Has the statistical analysis been performed appropriately and rigorously? 

Reviewer #1: Yes

4. Have the authors made all data underlying the findings in their manuscript fully available?

Reviewer #1: Yes

5. Is the manuscript presented in an intelligible fashion and written in standard English?

Reviewer #1: Yes

6. Review Comments to the Author

Reviewer #1: (No Response)

7. PLOS authors have the option to publish the peer review history of their article (what does this mean?). If published, this will include your full peer review and any attached files.

Reviewer #1: No

---

## [Author Response · Author response to Decision Letter 1]

17 Oct 2020

Dear PLOS ONE 

 Her is authors point by point response letter for the comments raised by the editor. The authors would like to thank the academic editor for his valuable comments. Care has been taken to improve the work and to address each specific comment. You can see the details of our modification under the track change version of the manuscript.

Additional Editor Comments (if provided)

Comment 1: Line 29 – suggest ‘involved interviews with each of these key stakeholders’

Response 1: Comment accepted and corrected. See under method section of the abstract and main manuscript.

Comment 2: Line 34 –The abstract needs some re-working for language. The results are presented as ‘x% agree/strongly agree with…’ doesn’t fit grammatically. I suggesting changing to “…agree/strongly agree that the trainers had adequate knowledge and skills….” Etc etc

Response 2: Comment accepted and corrected.

Comment 3: Line 50 – replace could with should

Response 3: accepted and word replaced. 

Comment 4: Line 109 – here it says 110 respondents – then in line 113 it says 107 - what happened to 3 participants?/which number is correct?

Response 4: Comment accepted and modified. Actually 110 is the calculated sample size for the study and 107 is the actual number of study participants. 

Comment 5: Line 113 – add ‘in-person’ and the location in which the interviews occurred

Response 5: Comment accepted and modified. See under data collection subsection.

Comment 6: Line 115 ‘close’ should be ‘closed’

Response 6: accepted and modified.

Comment 7: Lines 152 and 153 – I am unclear what you mean by diploma holders vs first degree professionals – can you please explain the qualifications of respondents?

Response 7: Comment accepted and modified. It is to mean laboratory technicians and BSc degree graduates in medical laboratory. See line number 154 under result section.

Comment 8: Line 164 – should be algorithm?

Response 8: It is to mean the new national HIV testing procedure. It is nationally agreed to be called “New National HIV Testing Algorism” training.

Comment 9: Table 2 – please just show the yes results. Being dichotomous, the ‘no’s’ can be assumed from this

Response 9: Comment accepted and the No response is deleted. See table 2 line number 172 and table 6 line number 226.

Comment 10: Line 205 – ‘as a potential to confident to the trainers’ – this needs to be re-worded

Response 10: Comment accepted and corrected.

Comment 11: Line 231 – “Agreeing with the training set-up” doesn’t make sense grammatically – was the question that it was appropriate/adequate/conducive to learning…please clarify

Response 11: Comment accepted and modified.

Comment 12: Table 8 talks about the ‘effectiveness’ of different elements – I suggest using ‘effective’ where describing the learning and transfer of knowledge to the workplace, but ‘appropriate’ or ‘adequate’ may be better for other elements such as the training set up

Response 12: Comment accepted and the words are modified. See table 8 line number 250

Comment 13: I was also interested in the comment about the judgment regarding access to perdiem – I think this needs some exploration as it is unclear why this would occur and in what context.

Response 13: Comment accepted and corrected. See under discussion section line number 291 and 292.

---

## [Editor Report · Decision Letter 2]

30 Oct 2020

PONE-D-20-10999R2

Evaluation of in-service training program of laboratory professionals in Amhara Public Health Institute Dessie Branch, northeast Ethiopia: a concurrent mixed-method study

PLOS ONE

Dear Dr. Hassen,

Thank you for submitting your manuscript to PLOS ONE. After careful consideration, we feel that it has merit but does not fully meet PLOS ONE’s publication criteria as it currently stands. Therefore, we invite you to submit a revised version of the manuscript that addresses the points raised during the review process.

Please engage copyediting to address the residual grammatical and language concerns with the paper.

We look forward to receiving your revised manuscript.

Kind regards,

Conor Gilligan

Academic Editor

PLOS ONE

Additional Editor Comments (if provided):

The authors have done a comprehensive job of addressing the previous comments but there remain problematic language and grammatical errors in the paper. I request that you pursue professional copyediting to enable further consideration of this manuscript.

---

## [Author Response · Author response to Decision Letter 2]

7 Nov 2020

Dear PLOS ONE 

 Her is authors response letter for the comments raised by the editor. You can see the details of our modification under the track change version of the manuscript.

Additional Editor Comments (if provided):

Comment 1: The authors have done a comprehensive job of addressing the previous comments but there remain problematic language and grammatical errors in the paper. I request that you pursue professional copyediting to enable further consideration of this manuscript.

Response 1: Comment accepted and corrected. The manuscript is edited with someone else who is fluent in English language. Besides, as you can see the track change version of the manuscript, care has been taken to improve the work and to address language problems by the authors.

---

## [Editor Report · Decision Letter 3]

17 Nov 2020

Evaluation of in-service training program of laboratory professionals in Amhara Public Health Institute Dessie Branch, northeast Ethiopia: a concurrent mixed-method study

PONE-D-20-10999R3

Dear Dr. Hassen,

We’re pleased to inform you that your manuscript has been judged scientifically suitable for publication and will be formally accepted for publication once it meets all outstanding technical requirements.

Kind regards,

Conor Gilligan

Academic Editor

PLOS ONE

Additional Editor Comments (optional):

Thank you for your effort to improve the English in your manuscript.
---

## [Editor Report · Acceptance letter]

23 Nov 2020

PONE-D-20-10999R3 

Evaluation of in-service training program of laboratory professionals in Amhara Public Health Institute Dessie Branch, northeast Ethiopia: a concurrent mixed-method study 

Dear Dr. Hassen:

I'm pleased to inform you that your manuscript has been deemed suitable for publication in PLOS ONE. Congratulations! Your manuscript is now with our production department. 

Kind regards, 

on behalf of

Dr. Conor Gilligan 

Academic Editor

PLOS ONE